# The Relationship between Social Media and the Increase in Mental Health Problems

**DOI:** 10.3390/ijerph20032383

**Published:** 2023-01-29

**Authors:** Hasan Beyari

**Affiliations:** Department of Administrative and Financial Sciences, Applied College, Umm Al-Qura University, Makkah 24382, Saudi Arabia; hmbeyari@uqu.edu.sa

**Keywords:** social media, mental health, analytical hierarchical process (AHP), followers, posts, Saudi Arabia

## Abstract

Social media has become an indispensable aspect of young people’s digital interactions, as they use it mostly for entertainment and communication purposes. Consequently, it has the potential to have both positive and negative effects on them. Deterioration in mental health is one of the side effects stemming from social media overuse. This study investigates the relationship between social media and the increase in mental health problems in Saudi Arabia. The population considered for analysis includes young people from Saudi Arabia, with a sample size of 385. A closed-ended survey questionnaire was used to collect data on different social media features and criteria. Using the Analytical Hierarchical Process (AHP), the researcher analyzed data to compare the effect of different social media features on mental health. The social media features included in this paper are private chats and calls, group chats and calls, browsing posts, games, media sharing, adverts, likes/comments/followers, and pages. The researcher adopted entertainment, information, social interaction, privacy, esteem, and communication as the criteria in the AHP process. Among these criteria, the study found that entertainment was the most significant, while privacy was the least significant. Findings suggested that likes, comments, and followers were the biggest contributors to poor mental health (total utility = 56.24). The least effective feature was ‘games’ (total utility = 2.56). The researcher recommends that social media users be cautious when interacting with social media features, especially likes, comments, followers, media, and posts, because of their significant effect on mental health.

## 1. Introduction

Mental health is a crucial aspect of human wellbeing, yet it is often overlooked and stigmatized. According to the World Health Organization, the prevalence of mental health problems is increasing at a rate of 13% per year [1]. Anxiety and depression are the most common mental health issues, affecting 264 million and 280 million people worldwide, respectively [2,3]. In addition, an estimated 269 million people were struggling with drug and substance abuse by the end of 2018 [4]. These numbers are likely to continue to rise due to a variety of factors. One factor that has been identified as contributing to the increase in mental health challenges is the use of technologies, including social media. Social media refers to applications that allow users to interact with each other through the creation and exchange of media, text, and calls within a network [5]. Some examples of social media platforms include Facebook, Twitter, Instagram, and TikTok. Key social media features considered in this investigation are private chats, group chats, browsing posts, adverts, media sharing, calls, likes and comments, and pages. Social media has been linked to poor sleep patterns, depression, and anxiety [6]. In addition, ref. [7] warns of the negative impact that excessive social media use can have on the mental health of young people.

Saudi Arabia has a high level of social media usage, with 82.3% of the population (29.5 million people) using social media in 2022 [8]. Young people, who make up 36.74% of the population, are the biggest users of social media in Saudi Arabia, with 98.43% of young people using social networking sites [9]. The top three reasons given by Saudis for using social media are keeping in touch with friends and family, use of free time, and finding products to purchase [8]. The prevalence of mental health issues in the KSA is estimated to be around 20.2% [10]. Depression is the most common mental health condition, affecting 21% of the population, followed by anxiety (17.5%) and stress (12.6%) [11]. Research has shown that social media use in Saudi Arabia is correlated with increased mental health issues [12]. High social media exposure has also been found to be associated with a higher risk of depression and anxiety in the kingdom [12]. Studies have also shown a significant correlation between the use of social networking sites and the increase in depression-related conditions in Saudi Arabia [13].

The aim of this study is to examine the impact of social media on mental health in Saudi Arabia and to identify which social media features have the greatest impact on increasing mental health issues. The study uses an Analytical Hierarchical Process (AHP) to analyze several social media features and determine their impacts on mental health. By understanding the specific features that contribute to mental health problems, individuals and policymakers can take steps to alleviate mental health issues and reduce the negative effects of social media. The results of this study will provide valuable insights into the impact of social media on mental health in Saudi Arabia and can inform the development of strategies to mitigate these effects.

## 2. Literature Review

One of the primary features of social media is chatting. As a social network, chats are a powerful method of communication among social media users. They may take the form of group or private chats. According to [14], young people with psychological issues tend to worsen their conditions by participating in social media chatrooms. Private chats are not exempted, as ref. [15] found that constant chatting with other people without feeling their physical presence is one reason for the increase in mental health issues among social media users. The outcome is more loneliness, a common factor in psychological deterioration. While chatting may not directly cause depression and other mental health problems, it can exacerbate an individual’s symptoms if one engages in long chats [16]. The studies further caution that young people must be careful when chatting with their peers on social media.

Browsing posts and advertisements are equally part of social media. Social media posts often portray falsehoods by allowing one to elevate their good qualities and suppress their negative ones [17]. Young people may not understand this fact, and they are likely to think that something is wrong with themselves because they do not look as good as the posts made by their friends. The authors of [18] found that social media influencers significantly contribute to the poor mental health of social media users. Advertisements power most social networking platforms, and users have had to embrace the presence of ads alongside their digital social lives. Because of their wide viewership, ads shape the psychology and opinions of young people on these platforms [19]. An advertisement portraying a muscular individual may depress a social media user who does not have similar body features. Similarly, ads with tall girls may negatively impact young girls psychologically because of social projection.

Sharing media, playing games on digital social networks, and interacting on video conferencing channels may negatively impact an individual’s mental health. In some cases, ref. [14] found that the sharing of media and interactions on social media prompts users to think less of themselves. Some users may not have good enough videos because their equipment, such as cameras, is not as good as their friends’ devices. Moreover, watching videos on social media can be an addictive habit if left unchecked. The authors of [20] argue that the active watching of and commenting on YouTube videos makes the platform overly addictive compared to people who passively watch videos without associated interactions. The authors advise that people’s interactions on video-based social media platforms should be minimal. Regarding games, ref. [21] argues that high involvement in social media games can result in addiction. Such a condition may make an individual overly dependent on these games, which distorts their mental health.

An individual’s following and the intensity with which people react to their posts can impact their mental health. For example, ref. [22] reports that users who update more frequently on their social media pages tend to receive more feedback in the form of likes and comments. This feedback is important, as it enhances the self-esteem of post authors. Moreover, ref. [23] observes that people receiving negative feedback from their social media posts are more susceptible to emotional distress. The study affirms that technologies aiding young people in comparing social statuses present a risk to their mental wellbeing. Some turn to social media to increase followers and gain a sense of gratification to compensate for their emotional and psychological challenges [24]. This leads them further down the path of a graver depression.

## 3. Methodology

This section provides an explanation of the methodological processes that the researcher used in order to acquire data and analyze them. The research design of this study is described in Section 3.1, which is then followed by the population, the sampling method, and the survey instrument. The phases of the Analytical Hierarchical Process (AHP) used in the research are explained in the following subsections.

### 3.1. Research Design

The specific approach taken by the researcher is the Analytical Hierarchical Process (AHP). It is a decision-making model that uses paired comparisons to determine the most significant factors that affect a decision [25]. In this case, the researcher wished to identify and rank social media factors impacting mental health. This ranking will help in prioritizing which aspects of social media use to manage at a personal level. The elements of social media in this study are private chats, group chats, browsing posts, adverts, media sharing, calls, likes and comments, and pages. The study undertakes a survey that asks respondents to indicate how useful these social media features are to them and how each element may lead to mental health problems.

### 3.2. Population, Sampling, and Survey Instrument

This study considered Saudi Arabia as the unit of study, while the study population was Saudi youth aged between 18 and 35. The United Nations defines youth as persons between 18 and 24. However, the researcher sought a more accommodating criterion regarding respondent ages. The selection of young people as the target population was motivated by the fact that 98.43% of them are on social media [9]. In addition, ref. [9] also reports that 7,623,336 young people belong to this demographic. The computed sample size from this population is 385 using Yamane’s formula [26]. Gender-wise, the researcher allowed respondents to indicate whether they were male, female, or non-binary. All respondents selected either the male or female category. Hence, the researcher analyzed the results in this fashion. The sample for this study was selected using simple random sampling on social media platforms such as Facebook and Twitter. This sampling method involves selecting participants randomly from the target population, which in this case were young people in Saudi Arabia who use social media. This helped to ensure that the sample was representative of the target population and that the responses were accurate and reliable. To ensure the content validity of the questionnaire, a pre-test of the survey was performed, since it is in the researcher’s best interest to have expert evaluations and reviews of the comprehensibility and clarity of the used research instrument. Several questions were altered, reworded, or eliminated in response to positive comments and ideas for small modifications. The amended questionnaire was forwarded to the collaborating academics for review and evaluation to confirm the instrument’s face validity. This questionnaire’s question types were determined by their degree of relevance to each identified concept. The Content Validity Index (CVI) was calculated to be 1, indicating that all three questions were relevant and appropriate for the study. This suggests that the questionnaire was valid and that it measured the variables of interest in a reliable and accurate manner.

The researcher used social media platforms to reach a diverse and representative sample of young people in the country. The social media platforms used in communication with participants (personal and business) included Facebook, Instagram, Twitter, and Snapchat. The researcher sent out a post including all the details about the research, and a link was included to direct the participants to the questionnaire page. The questionnaire was hosted on Google Forms to facilitate distribution, and it was left open for one month to allow respondents to respond at their convenience. The final questionnaire had a two-part structure, including demographic questions and three main questions with selective options for participants. Appendix A shows the list of questions asked to the respondents.

### 3.3. Analytical Hierarchical Process

The Analytical Hierarchical Process involves four primary steps, which are

Identifying decisions, options, and criteria;Conducting pairwise comparisons;Computing weights for the criteria;Calculating utility values.

#### 3.3.1. Identifying Decisions, Options, and Criteria

The decision is determining which social media features have the biggest effect on increasing mental health problems. The options were the eight social media features, namely private chats, group chats, browsing posts, adverts, media sharing, calls, likes and comments, and pages. The criteria for determining which features are the most influential were the importance of a feature to an individual, the time spent interacting with the feature, and the recency of interaction.

#### 3.3.2. Pairwise Comparison

Pairwise comparisons involve comparing two criteria simultaneously to build a square n × n matrix, where n is the number of criteria. The comparison is structured in such a way that the value entered in a cell represents the number of times one criterion is more important relative to the other. Because the two criteria being compared are the same, the relative value of each criterion is equal to one when they are compared to each other [25]. The maximum possible score is n, and larger numbers indicate that a criterion is becoming essential. The pairwise comparison will compare time spent on a feature, recency in using the feature, and the overall importance of the feature to the respondents.

#### 3.3.3. Importance Weights

After populating the matrix, it is used to compute the importance weights. They signal to an analyst the extent to which each criterion will affect their ultimate decision. The researcher gave the biggest weight to the item with the most significant importance. The study computed the geometric mean of the criteria to ensure objectivity in the computation in the first step, as suggested by [27]. In the second step, the relative composition of the criterion values was determined, which was used to determine their weights [28]. In order to complete the procedure, the computation of the ratio of the value of each criterion to the overall value is needed.

#### 3.3.4. Calculating Utility Values

Computing the utility is the final step in the analytical hierarchal process. It involves establishing the ‘utiles’ associated and multiplying them by their corresponding importance scores [27]. The ‘utiles’ are obtained using respondents’ subjective evaluation of how each feature instigates mental health challenges. ‘Utility’ is a quantitative value that indicates how useful something is to an individual. This figure helps in selecting the most significant option. It is possible to represent utility as a percentage. It is argued that a criterion’s usefulness increases as its advantages or benefits increase. Depending on the criterion, it is conceivable that utility will be computed differently. The importance of the criteria selected for investigation and the utility attached to the criterion were multiplied to show the utility calculation for each criterion. The values for each criterion were added to determine the total utility of each social media feature.

## 4. Results

### 4.1. Analysis of Demographic Characteristics

This section analyzes the age, gender, and occupations of the study participants. The findings reveal that the most populous age group was that of members aged between 18 and 25, as they constituted 60.3% (232) of the study population. Male respondents accounted for 55.3% (213) of the sampled participants. The most dominant group by occupation was students, as they accounted for 41.8% (161) of the sampled participants. Table 1 provides further details about the demographic characteristics of the respondents.

### 4.2. Favorite Features of Respondents

The researcher first examined which of the selected social media features were favored by the respondents. The findings suggested that likes, comments, and followers were the most relevant aspects of social media that the respondents liked, obtaining a mean score of 7.29/8.00. The least favorite feature was gaming, scoring a mean of 2.05/8.00. Table 2 shows the performance of the different features.

### 4.3. Pairwise Comparison

The researcher established the criteria comparison matrix using the responses to questions that asked participants to rank the factors influencing their sentiments on social media features. The ranking was based on the mean score obtained from the 385 responses regarding their criteria ranking. In this case, the highest ranked criteria by the respondents scored higher values in Table 3. Evidence suggests that people decided which social media feature they valued mostly based on its entertainment value (value = 6) and less so based on the feature’s privacy (value = 1).

The computation of matrix values in Table 4 was based on the values established in Table 3 above. The basis of the values is the mean ranks of the criteria, as expressed by the respondents. In this case, the matrix values indicated the number of times one criterion was more important than the corresponding criterion [28]. For example, the highlighted pair in Table 4 shows that esteem was two times more important that the corresponding information criterion.

### 4.4. Importance Weights

The first step involves the computation of the criteria’s geometric mean [28] to determine their influence on the final decision. In this case, it is the sixth root of the product of the row elements in Table 4. Below is the basic formula used in computing the weights of the criteria, assuming n criteria:$$V_{i}=\sqrt[{n}]{X_{i1}\times X_{i2}\times \cdots \times X_{in}}$$
where:*V_i_*: Geometric mean for criterion *i*;*X_i_*_1_: Pairwise importance of criterion *i* relative to criterion 1;*X_i_*_2_: Pairwise importance of criterion *i* relative to criterion 2;*X_in_*: Pairwise importance of criterion *i* relative to criterion *n*;*n*: Number of criteria.

The second step involves finding the proportionate composition of the criteria values, which will count as their weights [28]. The procedure requires the computation of the ratio of each criterion’s value against the total value:$$W_{i}=\frac{V_{i}}{\sum_{j=1}^{n}V_{j}}$$
$$\sum_{j=1}^{6}V_{j}=7.014$$
where:*W_i_*: Weights for criterion *i*.

### 4.5. Computing Utility Values

The researcher computed the feature utiles by first ranking their respective mean responses. The findings in Table 5 show that respondents thought that likes, comments, and followers on social media would often cause people’s mental health problems. Other similarly high-risk features are browsing posts and adverts.

### 4.6. Comparing Social Media’s Effects on Mental Health

This study computed the total utility as the product of the utiles (feature strengths), importance weights (criteria weights), and how favored the features were by the respondents (relevance). In Table 6, each feature’s strength is multiplied by the criteria weights to obtain the cell values. The row values are then added and multiplied by a feature’s importance to determine the total utility. The total utility is obtained using the following formula:$$TU_{i}=\overset{n}{\underset{j=1}{\sum }}W_{i}\times UV_{j}\times MR_{i}$$
where:*TU_i_*: Total Utility for criterion *i*;*W_i_*: Weights for criterion *i*;*UV_j_* = Utility Value for feature *j*;*MR_i_*: Mean Relevance for criterion *i*;*i* from 1 to 8, *j* from 1 to 6.

**Table 6 ijerph-20-02383-t006:** Estimating the effect of social media features on mental health problems.

			Criterion Weights							
			0.29	0.10	0.24	0.05	0.19	0.14		
			ENT	INF	SOC	PRI	EST	COM	Mean Relevance	Total Utility
Feature Strength (Utility Value)	7.71	LCF	2.20	0.73	1.84	0.37	1.47	1.10	7.29	56.24416
	7.11	BRP	2.03	0.68	1.69	0.34	1.35	1.02	6.33	45.03454
	3.55	MDS	1.01	0.34	0.84	0.17	0.68	0.51	7.16	25.39835
	3.48	GCC	1.00	0.33	0.83	0.17	0.66	0.50	4.80	16.72801
	4.89	PGS	1.40	0.47	1.16	0.23	0.93	0.70	3.11	15.20282
	2.26	PCC	0.65	0.22	0.54	0.11	0.43	0.32	3.98	9.024443
	5.75	ADV	1.64	0.55	1.37	0.27	1.09	0.82	1.26	7.241052
	1.25	GMS	0.36	0.12	0.30	0.06	0.24	0.18	2.05	2.561511

The findings suggest that the feature with the most significant negative effect on mental health is ‘likes, comments, and followers.’ This feature scored a total utility of 56.24. On the other hand, the feature with the least significant negative effect on mental health is ‘social media games’. This study found the feature to have a total utility of 2.56. While the respondents had opined in Table 3 that adverts substantially contribute to mental instability, the criteria weights for this feature were too low to significantly impact the feature’s total utility.

## 5. Discussion

In this study, the researcher found that social media has a significant negative impact on the mental health of Saudi Arabian youth. The feature that had the greatest impact was likes, comments, and followers, with a utility value of 56.24. This suggests that individuals who are seeking validation and social esteem through social media may be more prone to experiencing stress, depression, and anxiety. Browsing posts and media sharing were also identified as significant features that negatively impact mental health, with utility values of 45.03 and 25.40, respectively. These findings align with previous research that has identified the presence of influencers on social media as a potential source of stress and depression for regular users who may feel pressure to emulate these individuals [18]. Additionally, excessive exposure to social media videos has been linked to negative mental health outcomes [20].

On the other hand, this study found that social media games had the least impact on mental health, with a utility value of only 2.05. This finding differs from previous research that has identified games on social media as highly addictive and potentially harmful to mental health [21]. However, it is important to note that this study only compared the negative impact of different social media features on mental health, and it is possible that social media games may have a greater impact when studied in isolation. These findings highlight the need for caution in the use of social media, particularly among young people in Saudi Arabia. While social media can provide a sense of connection and support, it is important to be aware of its potential negative impacts on mental health. In light of these findings, it may be beneficial for individuals to set limits on their social media use and prioritize activities that promote mental wellbeing, such as physical exercise and social interaction with friends and family.

One potential implication of these findings is the need for greater education and awareness about the potential dangers of social media. This could involve educating people about the importance of finding validation from sources other than social media, as well as helping people to develop healthy habits when it comes to their social media use. This could involve setting limits on the amount of time spent on social media, being selective about the content that is consumed, and finding ways to disconnect from social media when necessary. Overall, these findings highlight the need for caution when using social media, particularly for youth in Saudi Arabia. While social media can be a useful tool for communication and connection, it is important to be mindful of the potential negative effects on mental health. It may be helpful for individuals to limit the attention they pay to certain features, such as likes, comments, and followers, and to engage in passive rather than active consumption of media. Further research is needed to understand the specific mechanisms by which social media impacts mental health and to identify effective interventions to mitigate negative effects.

There are several potential limitations to this study that should be considered when interpreting the results. First, the sample size of 385 participants may not be representative of the larger population of Saudi Arabian youth. Additionally, the self-reported nature of the data may be subject to bias, as individuals may not accurately recall or report their social media habits. Finally, the cross-sectional design of the study means that it is not possible to establish cause-and-effect relationships between social media use and mental health. Another limitation of this study is that the definition of “youth” is not explicitly stated. It is possible that the experiences and activities of respondents aged 18 and those aged 35 may differ significantly. Additionally, the study did not explicitly consider the potential impact of gender on the relationship between social media use and mental health. Future research should aim to further explore these demographic variables in order to better understand the specific effects of social media on mental health among different populations. Such investigations should consider using larger and more diverse samples, as well as more robust research designs to further explore the relationship between social media and mental health.

## 6. Conclusions

The purpose of this study was to examine the effects of social media on mental health among young people. Social media has become an integral part of modern society, with platforms such as Facebook, Twitter, and Instagram offering a range of features including messaging, media sharing, and gaming. However, there is growing concern that the use of social media may have negative effects on mental health, particularly among young people who are more likely to use these platforms extensively. The study aimed to identify the specific features of social media that have the greatest impact on mental health and to examine the underlying reasons for these effects. To achieve these objectives, the study used AHP to assess the relevance and importance of eight social media features to 385 respondents aged between 18 and 35. The findings showed that likes, comments, and followers were the most relevant features to respondents, while gaming was the least favorite feature. In terms of the criteria influencing the respondents’ sentiments, entertainment was the most important factor, while privacy was the least important. The study concludes that social media can have both positive and negative effects on mental health, depending on how it is used and the specific features that are engaged with. It is therefore important for young people to be aware of the potential risks and to use social media in a balanced and responsible manner.

## Figures and Tables

**Table 1 ijerph-20-02383-t001:** Respondents’ demographic characteristics.

Demographics		Frequency	Percentage (%)
Gender			
	Male	213	55.3
	Female	172	44.7
Age			
	18–25	232	60.3
	26–30	114	29.6
	31–35	39	10.1
Occupation			
	Student	161	41.8
	Unemployed	138	35.8
	Employed	86	22.3
	Total	385	100%

**Table 2 ijerph-20-02383-t002:** Ranking the relevance of social media features to respondents.

Feature	Mean Relevance
Likes, Comments, and Followers	7.29
Media Sharing and Consuming	7.16
Browsing Posts	6.33
Group Chats and Calls	4.80
Private Chats and Calls	3.98
Pages	3.11
Games	2.05
Adverts	1.26

**Table 3 ijerph-20-02383-t003:** Criteria importance.

Key	Feature	Value
ENT	Entertainment	6
INF	Information	2
SOC	Social Interaction	5
PRI	Privacy	1
EST	Esteem	4
COM	Communication	3

**Table 4 ijerph-20-02383-t004:** Pairwise comparison matrix.

	Ranks →	6	2	5	1	4	3		
Ranks ↓		ENT	INF	SOC	PRI	EST	COM	V	W
6	ENT	1.00	3.00	1.20	6.00	1.50	2.00	2.004	0.28571
2	INF	0.33	1.00	0.40	2.00	0.50	0.67	0.668	0.09524
5	SOC	0.83	2.50	1.00	5.00	1.25	1.67	1.670	0.23810
1	PRI	0.17	0.50	0.20	1.00	0.25	0.33	0.334	0.04762
4	EST	0.67	2.00	0.80	4.00	1.00	1.33	1.336	0.19048
3	COM	0.50	1.50	0.60	3.00	0.75	1.00	1.002	0.14286

**Table 5 ijerph-20-02383-t005:** Utility values.

Feature	Utiles
Private Chats and Calls	2.26
Group Chats and Calls	3.48
Browsing Posts	7.11
Games	1.25
Media Sharing and Consuming	3.55
Adverts	5.75
Likes, Comments, and Followers	7.71
Pages	4.89

## Data Availability

The data presented in this study are available on request from the corresponding author.

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
