# Peer review of "The Relationship between Social Media and the Increase in Mental Health Problems"

_ijerph, 2023, doi:10.3390/ijerph20032383_

Round 1

Reviewer 1 Report (Previous Reviewer 2)

Dear authors.

The authors have improved the manuscript. Most of the suggestions in the previous review were done, but I still maintain some of the previous considerations concerning the methodology.

The methodology used is still presented in a very shallow way, which is not in line with this type of research where the methodological process is very relevant. Because of that, I reinforce again the recommendations done in the previous review:

(i) Who made the face and validity content of the questionnaire? Explain in detail;

(ii) How was the distribution of the questionnaire made? “The researcher used social media platforms to reach a diverse and representative sample of young people in the country” is not clear nor acceptable. Explain in detail, for example, the social networks (SN) used, the addresses of the SN, personal or business SN, and other relevant information concerning the distribution.   

(iii) How long was the questionnaire open? “…it was left open for one week…” This is too vague. Indicate the real period of time.

(iv) Made a brief description of the structure of the questionnaire.

See the “Figure 1. Development flow diagram of the questionnaire” of the manuscript “How Web 3.0 Tourism Students See the 1.0 Higher Education System”, to help with the methodological part of the questionnaire.

It is my recommendation, in addition to the manuscript, as supplementary material, to add the survey used.

Author Response

Dear Reviewer ,

Please see attchad file.

Regards,

Reviewer 2 Report (New Reviewer)

The paper claims to examine the relationship between social media and mental health problems. The paper does this by asking respondents to rate various aspects of social media in terms of their preferences and the impact to their mental health. This is not an appropriate method to answer the study's research question. Generally, when conducting a survey, we don't ask respondents directly about their mental health or what aspect of social media affects their mental health. Rather, we use validated mental health instruments and then test the correlation between an aspect of social media and the mental health issue that we are interested in. The paper also writes the findings as if the paper has proved causation. The author has thus confounded causation with correlation. 

Because of this significant methodological issue, the paper does not address its key research question (which is also too broad). 

Author Response

Dear Reviewer 2,

Regards,

Reviewer 3 Report (New Reviewer)

The article presents a valuable contribution for researchers who want to study the relationship between social media and the mental health.

The manuscript is well structured and presents very clearly and supported by scientific evidence, how different social media has been linked to poor sleep patterns, depression, and anxiety and the negative impact that excessive social media use can have on the mental health of young people.

To achieve this, aim the research used a calculated sample of 385 individuals to analyze the impact of social networks on mental health and to examine the underlying reasons for these effects.

With regard to the methodology, it is clear the research design, using Analytical Hierarchical Process (AHP), moreover the Analytical Hierarchical Process is explaining step-by-step, and the reason why this method was chosen for the research.

The results were well explored.

The discussion of the data is well grounded in the theory and evidence referred to in the rationale. On the other hand, it also presents some potential implications, and I am of the opinion that here, it could also pave the way to help the mental health area more from the perspective of health professionals, in order to be more aware of how social networks are used and what affects their users the most.

The limitations presented are relevant, and the way they pave the way for future research considering other socio-demographic variables.

Author Response

Dear Reviewer 3,

Regards,

This manuscript is a resubmission of an earlier submission. The following is a list of the peer review reports and author responses from that submission.

Round 1

Reviewer 1 Report

Thank you for the opportunity to read your paper. As you will see I have a considerable number of concerns about its content, structure and arguments as detailed below. I do feel that the topic is a worthy one, and would, suitably refined, make for a valuable addition to the literature. However, in the manuscript’s current state, I do not believe it is as of yet suitable for publication. I do hope you will be able to take these comments and use them to reconstruct the piece.

Abstract: ‘Social media has become an indispensable aspect of youths’ digital interactions, as they use it mostly for entertainment and communication purposes. Consequently, it has had positive and negative effects on them.’ Why consequently? This is unclear – is the author trying to suggest that exposure to entertainment or the ability to communicate with others is detrimental to their mental health? This seems a concerning, and possibly blinkered, statement which will strongly colour the perceptions of anyone reading the paper. Introducing an element of critical thinking here – e.g. ‘Consequently, it has the potential to have positive…’ would serve to assuage these concerns.

Syntax and grammar: I found the paper hard to read in many places and have picked out a handful of issues as examples below. The paper clearly needs further proofreading and refinement of language ahead of acceptance for publication.

Scholarly Voice/Scholarship: The paper as a whole reads and flows closer to a student assignment than an academic text. It presents ‘facts’ which are usually supported by a single reference. It does engage to a sufficient critical depth, nor offer counterpoints which makes for a more polemical than analytical presentation. The author would do well to also consider introducing more of their own insights and critique within the opening pages as it is difficult to adjudge if they are taking the prior literature at its word, without introducing their own analytical thought/insights. This is to the detriment of the paper, as I suspect the author has many valuable and insightful comments they could add to the narrative.

The paper is also written in a highly staccato form, with series of short statements following one after another, making any narrative flow or line of argument from the author problematic to follow. Attention needs to be deployed in creating a narrative flow, and improving the signposting within the work to allow readers to follow its logical progression. See for example lines 81-90 for an example of this staccato practice – although it occurs systematically throughout the paper.

Introduction: The statement of ‘improving mental health’ – is this from the perspective of the general public or therapeutic practitioners? In either case I would disagree with the axiom that there’s little focus on its improvement in both the literature and the public sphere. Post-covid/lockdown restrictions, there’s been if anything an upswell of attention in maintaining/establishing good mental health. Consequently, I think the first paragraph needs to be restructured as its opening sweeping statement does not align globally. However, if the author is writing solely from a single country perspective (e.g. Saudi Arabia), and hence the local climate is one where a disregard for improving mental health may be a truism.

The statements (line 40-49) that support the argument how ‘technologies have contributed to the surge in mental health challenges’ while bold, appear to be under-supported by the wider literature. I would have expected a broader and deeper reading, from a global perspective to support this claim: which while certainly credible, is presented a less than critical manner here. I would have expected literature counter to this position to also be introduced here and the author to counter it with their own insights.

As a whole then the introduction needs extensive revision, tightening of phraseology and supporting of currently unsupported statements. It does not make for a strong introduction to the paper, and would likely discourage the reader from continuing on to the main body of the work.

Definitions: The term ‘social media’ is broad and ill-defined, even within the wider literature. Hence, it is vital The author should make efforts to indicate what they include and exclude. E.g. Facebook, blogs, Instagram, SnapChat, Open Journal review (for example) – all form part of a social media digital landscape but do not operate nor are their user bases similar. WhatsApp is cited as social media but falls down on many standard social media criteria as it is primary a messaging app, rather than a public (e.g. social) interaction space/platform. The author should take the space to demarcate the landscape and definitions as to ‘what is/isn’t’ social media and hence the borders within which their analysis and arguments lie, as currently this is absent from the work.

Methodology: The introduction to the section, and subsequent paragraph repeat the approach used, making elements of each redundant. Refining these sections to read more logically would be advised.

Discussion: The discussion, following a sweeping statement (which feels distinctly unearned and would be better at the end or in the conclusion following a robust discussion of the findings) based on the study’s findings – rather than exploring the author’s thoughts, critique and insights from their analysis instead introduces a large amount of information from no fewer than 5 other sources. These should have been included in the earlier literature review, with the discussion space given over more or less entirely to the authors original thought. As such while the empirical results are original, the crucial discussion is largely predicated on prior work – and fails to introduce sufficiently original work or thinking from the author. Along with the issues discussed above in the introduction, this section is the other major failing of the paper and should be entirely rewritten to produce the analytical critique and development of themes elucidated from the original research survey work.

Conclusion: This is relatively well written, but based on the prior discussion the validity of these findings is unclear. The author is also speaking very generally and it is important to reiterate here that these findings are based on a specific demographic from within a specific country and as such cannot be generally applied. However, some mention of testing the validity with other (external) populations or different age/gender demographics might make for valuable follow on work.

Some minor points/examples of syntax concern.

29-31: ‘wellbeing’ should be unhyphenated.

34: ‘264 million victims’ – victims is an extremely pejorative term and should be replaced with ‘sufferers’ or a similar.

35-36: ‘These statistics keep worsening 35 with time because of several factors that keep emerging with time.’ – this is a very clunky sentence (the use of time twice for example) and needs rephrasing.

37: ‘mental health problems grows at’ – suggest syntax would be improved with ‘…problems are increasing at…’

38-39: ‘While treating the conditions remains the focus of health organizations, a proper understanding of the causes can help minimize the incidence of these conditions.’ – again a clunky sentence (cf. use of conditions twice). Additionally, seems to disagree with the author’s own opening statement about the lack of focus on improving mental health wellbeing. If it IS a focus for health organisations, how can there be a minimal focus on improvements?

48 ‘[8] caution against youths’ overindulgence…’ – better phrased as ‘Some authors [8] also caution against…’

61: ‘Saudi Arabia’s state of mental health is average.’ – this is an unsupported and highly questionable statement, which seems at first reading to be a potential example of author cultural bias. This needs support from the prior literature (especially that written from an outsider perspective) and an expansion on what and how ‘average’ is defined.

62-63: ‘The most significant condition is depression, as it accounts for at least 21% of the population.’ Poorly phrased – would be much improved as ‘The most significant and frequently reported condition within the country is depression, which one study demonstrated affects around 21% of the population’. – that is assuming reference [2] supports this statement. If not, a further study should be cited. The next statement on anxiety is similarly weakly constructed and needs revision for clarity.

83: ‘psychotic issues’ I believe is an archaic, outmoded and déclassé term. Please replace with modern accepted parlance.

132: Duplicate, unnecessary acronym explanation (AHP)

135-139: This whole paragraph needs to be rewritten to be clearer narratively. Additionally, the elements of social media would have been better in the general introduction. (cf. comments on social media definitions).  

142: How is ‘youth’ defined – is this a standard definition or one of the author? A demographic of 18-34 seems broadly reductive when the experiences, life and activities of an 18 year old are incomparable with those in their 30s. Additionally, no evidence of considerations of gender appear, which are an essential component of any population based study – again experiences of different sexes (including non-binary affiliation) cannot and should not be normalised as part of a homogeneous block. I note with satisfaction that the author does account for at least the two ‘normative’ genders within their analysis later – but more attention/exploration of their demographic base is needed here.

142-143: Repeated, redundant statement (see: line 53)

Author Response

Dear Reviewer 1,

Please see attached files including the responses to the comments and the final paper with all required changes.  

Kind regards,

Reviewer 2 Report

Dear authors,

This is an interesting manuscript that analyses the relationship between social media and the increase in mental health problems.

However, the manuscript has important flaws namely:

.  Review the citations [2] and [3] in the sentences: (i) “Anxiety is the most common mental health problem, affecting 284 million people worldwide [2].”, and “It is closely followed by depression, whose estimated prevalence is 264 million victims, while those undergoing substance abuse issues are about 71 million [3]” because both papers do not have any information concerning the data on the sentences;

. All the external data in the sections “Introduction”, and “Literature Review” must have a citation. See these three examples where the citations are missing: (i) “The proportion of Saudis suffering from mental health conditions is estimated to be between 15 and 20%.”; (ii) “The most significant condition is depression, as it accounts for at least 21% of the population.”, and (iii) “Anxiety and stress account for 17.5% and 12.6%, respectively.”.

. Also, these three sentences must have a citation: (i) “Social media posts often portray falsehoods by elevating one’s good qualities and suppressing their negative ones.”; (ii) “Such a feeling can be disastrous to young people, who may think their parents do not love them as much.”, and (iii) “In this study, the researcher used Yamane's method to determine that a sample size of 385 was necessary.”.

Despite the questionnaire being designed from the former literature it must be better described:

(a) Was the questionnaire's face and validity content assessed? If yes, by who?;

(b) How was the sample gathered? How were the respondents selected? Is it a random sample?

(c) How was the distribution of the questionnaire made?

(d) How long was the questionnaire open?

It is my recommendation, in addition to the manuscript, as supplementary material, to add the survey used.

Given the content of the manuscript, and the reasons mentioned above (research not conducted correctly), I will reject, at this time, the manuscript. However, I believe this research restructured has the potential to be published in the nearby future.

Author Response

Dear Reviewer,

Please see attached files including responses to the comments and the final version of the paper.
